# Development of Ceramic Tiles from Philippine Nickel Laterite Mine Waste by Ceramic Casting Method

Ivyleen C. Bernardo-Arugay [1,*], Fel Jane A. Echavez [1], Rae Homer L. Aquiatan [1], Carlito B. Tabelin [2], Raymond V. Rivera Virtudazo [1] and Vannie Joy T. Resabal [1]

1   Department of Materials and Resources Engineering Technology, Mindanao State University—Iligan Institute of Technology, Tibanga, Iligan City 9200, Philippines; feljane.echavez@g.msuiit.edu.ph (F.J.A.E.); raehomer.aquiatan@g.msuiit.edu.ph (R.H.L.A.); raymondrivera.virtudazo@g.msuiit.edu.ph (R.V.R.V.); vanniejoy.resabal@g.msuiit.edu.ph (V.J.T.R.)
2   School of Minerals and Energy Resources Engineering, University of New South Wales, Sydney, NSW 2052, Australia; c.tabelin@unsw.edu.au
*   Correspondence: ivyleen.arugay@g.msuiit.edu.ph

**Abstract:** Siltation is the biggest environmental challenge associated with nickel laterite mining in the Philippines. The amount of silt generated is huge and one mitigation strategy currently employed by the mining companies is the construction of siltation ponds where the bulk of the clayey- and silt-sized surface runoffs is collected. However, this poses several serious environmental hazards such as landslides due to heavy rainfall and the potential release of hazardous heavy metals. A promising approach to reduce the risks associated with long-term storage of nickel mine waste (NMW) is to employ circular economy by repurposing it for ceramic applications. While generating useful materials with economic value out of a mine waste, it will result in a reduction in volume of waste for disposal. In this study, the method employed to produce NMW-based ceramic wall and floor tiles is slip casting as it is the most appropriate method in forming tiles with complex surface features. Five formulations of NMW-based slips were made for the casting of ceramic tiles and each slip was characterized for its suitability as raw material. The results of NMW characterization show that NMW could be utilized as raw materials for both ceramic wall and floor tiles and the addition of feldspar can enhance casting and physical properties.

**Keywords:** nickel mine waste; siltation pond; repurposing of waste; ceramic tiles; circular economy; Philippines nickel mine; ceramic casting; plasticity; deflocculation; waste characterisation

## 1. Introduction

Climate change is one of the most serious and devastating problems facing humanity this century. The global mean temperature in 2020 was around $1.2 \pm 0.1\ ^\circ$C higher than pre-industrial revolution levels (1850–1900). This temperature increase was largely attributed to the release of anthropogenic greenhouse gases (GHGs) in the atmosphere such as carbon dioxide ($CO_2$) and methane ($CH_4$) [1]. To combat climate change, 196 parties ratified the United Nations Sustainable Development Goals (UN-SDGs) #13 "Climate action" with the goal of a global $CO_2$ emission reduction of 45% from 2010 levels by 2030 and achieve net-zero emissions by 2050. The bulk of GHG emissions is directly linked to the burning of fossil fuels—coal, oil and natural gas—to satisfy the energy needs of our modern society. In 2018, for example, ~50 billion tons (t) of $CO_2$ was released globally, and the five largest emitters were the electricity and heat generation sector (31%), transport sector (16%), manufacturing and construction (12%), agriculture (12%) and industry (6%) [2]. This means that replacing fossil fuels with renewable energy and clean energy technologies in the electricity/heat generation and transport sectors could translate to substantial reductions in $CO_2$ emissions.

Unfortunately, renewable energy and clean storage technologies are more metal, mineral and material intensive compared with conventional fossil fuel-based technologies. For

example, electric vehicles (EVs) require 1.7 to 11 times more copper (Cu) than conventional cars in addition to critical elements such as cobalt (Co), nickel (Ni), manganese (Mn) and lithium (Li), which are important in rechargeable batteries [3]. According to a recent report of the World Bank, the supplies of 17 metals/materials—aluminum, chromium, Co, Cu, graphite, indium, iron, lead, Li, Mn, molybdenum, neodymium, Ni, silver, titanium, vanadium and zinc—should be stable in the next 30–50 years for clean energy transition to succeed [4]. Thus, mining for critical elements will become even more extensive to cope with the demands of these metals/materials worldwide.

The Philippines is the world's fifth most mineral-rich country with resource assets estimated at USD 1 trillion in untapped metallic reserves [5] and a major player in the nickel supply chain. According to the United State Geological Survey (USGS, Reston, VA, USA), the Philippines has 4.8 Mt of Ni resources as of 2020. The country ranks sixth in the world and accounted for 5% of global Ni reserves. It is also the second largest nickel producer in 2019 and 2020 supplying 323,000 and 320,000 t of Ni, respectively [6]. In addition to Ni, laterite deposits in the Philippines have been reported to contain Co [7,8], another critical element important in the manufacture of cathodes for rechargeable batteries [9].

The 29 operating nickel laterite mines in the Philippines [10] employ the conventional open pit (i.e., excavation and haulage) mining method. Because most of these mines do not have in-house hydrometallurgical operations for Ni extraction and the ore is typically sent to China for further processing, siltation is the biggest environmental challenge associated with Ni laterite mining in the Philippines. One mitigation strategy currently employed is the construction of "siltation ponds" in key areas of the mine to capture the bulk of clayey and silt-load surface runoffs, allowing fine particles to settle via sedimentation. The overflows from these siltation ponds are controlled to limit the negative impacts of siltation to surrounding rivers or aquatic bodies, farmlands and coastal areas. To keep the siltation ponds at optimum conditions, they are regularly excavated and/or dredged and the excavated/dredged waste materials—referred to as nickel mine waste (NMW) in this paper—are disposed of in embankment-type repositories that are rehabilitated by covering with productive soils and replanting.

The amount of NMW generated by Ni laterite mining operations is huge. In Agusan, Philippines, for example, one of the mine sites produced voluminous silt at around 74,800 $m^3$, 252,500 $m^3$ and 80,900 $m^3$ for the years of 2016, 2017 and 2018, respectively. Although the use of siltation ponds in Ni laterite mines is effective, the NMW management approach is unsustainable. The cyclic dredging and hauling of NMW to rehabilitation sites from siltation ponds entails at least high fuel consumption and manpower costs. Because NMW are simply "stored" on site, these waste embankments will continue to pose serious safety and environmental risks similar to those associated with tailings storage facilities (TSFs) and waste dumps [11,12]. Landslides or mudslides, for example, are potential hazards in the Philippines due to heavy rainfall during the monsoon season or earthquake-induced liquefaction. The release of hazardous heavy metals such as chromium is another hazard associated with NMW, especially when exposed to reducing-oxidizing cycles prevalent in replanted and rehabilitated waste dumps [13].

One alternative and promising approach to reduce the risks associated with long-term storage of NMW is to repurpose this waste for ceramic applications. Employing a circular economy approach to manage NMW has three advantages [14]: (i) reduce the volume of waste for disposal, (ii) generate useful materials with economic value and (iii) cultivate and support the local ceramics industry. The high potential of NMW for ceramic applications is related at least to its abundant finely sized material content, including its chemical and mineralogical properties. The NMW generated in laterite Ni mines is dominated by silt-sized (0.2–63 μm) to clayey-sized (<0.2 μm) particles, which is a direct consequence of particle segregation in flowing water. In terms of mineralogy, the host rock of Ni laterite deposits in the Philippines are composed of iron oxides (e.g., goethite and hematite), silicates (e.g., quartz and chlorite-group) and clay minerals (e.g., talc and serpentine–kaolinite group) [11,12], which are found in raw materials used by the ceramics industry.

In this study, NMW from one of the siltation ponds of a Ni laterite mine in the Philippines was characterized and repurposed to produce wall and floor tiles. The volume of NMW in this pond was approximately 40,495 m$^3$ and 17,989 m$^3$ for the years of 2019 and 2020, respectively. To the best of our knowledge, this is the first work to report the potential of NMW from the Philippines as partial or full replacements for the clay minerals such as kaolin, bentonites and typical red clay for the production of wall and floor tiles. Slip casting method of NMW-based ceramic slips was employed to form the tiles because this method is the most appropriate in forming tiles with complex surface features in contrast to plain and smooth surface, since these features are difficult to achieve using other forming methods such as wet-pressing and dry pressing. The success of slip casting tiles with complex surface features will also lead to the casting of other ceramics with similar complex surface features. The development of ceramic tiles from NMW involved the following: (i) evaluation of the particle size distribution of NMW, (ii) determination of the chemical composition and mineralogical properties of NMW, (iii) evaluation of the casting properties of pure and formulated NMW slips, and (iv) characterization of the physical properties of NMW tiles fired at different temperatures.

## 2. Materials and Methods

### 2.1. Sample Preparation of the Nickel Laterite Mine Waste

The NMW samples were collected from siltation ponds of a mining company in Agusan del Norte, Northeastern Mindanao, Philippines. The "lumpy" and "soggy" samples were sun dried to remove the moisture content and then crushed using a UA V-Belt Drive pulverizer, (BICO Braun International, Burbank, CA, USA) with a power of 3HP, 8 inch grinding plates and speed of 900 rpm. Commercial feldspar and technical sodium silicate that will be added to the NMW samples for the preparation of the green bodies were bought from a local supplier in Mindanao, Philippines. The particle size distribution (PSD) of commercial feldspar is less than 74 μm, which means that no further grinding is necessary.

### 2.2. Raw Material Characterization

The PSD analysis of NMW was performed by wet screening using Tyler standard sieves following the sieve size fraction series: 425 μm, 212 μm, 150 μm, 106 μm, 74 μm, 44 μm and 37 μm. To obtain the residue at each size fraction, tap water is continuously poured onto the sample in the sieve until the water passing through the sieve becomes clear. The residues at each size fraction and the materials that went through the smallest size fraction, i.e., 37 μm, were separately dried and then weighed.

The chemical composition of the NMW sample was determined using XRF (Olympus Innov-X Pro X-ray Fluorescence Spectrometer, (Olympus Innov S, Woburn, MA, USA) while mineralogical analyses were conducted by X-ray Powder Diffraction where data collection is performed using a Rigaku Miniflex 600, (Rigaku, Tokyo, Japan) with CuKα radiation source ($\lambda = 3.541838$) and a measuring angle of 5° to 75° (2θ) with operative conditions of 40 kV voltage and 30 mA current. Results of the X-ray Powder Diffraction of NMW were primarily based on "figure-of-merit" values from available references [15–17].

To examine the suitability of the NMW as raw material for ceramic tiles, its plastic behavior must be evaluated through the determination of the Atterberg limits, which involves the measurement of the liquid limit and plastic limit to obtain the plasticity index. The method used for the determination of Atterberg limits is in accordance with ASTM D4318-10 [18–20].

The thermal stability of the NMW as raw material for ceramic tiles was determined by performing Thermogravimetric and Differential Thermal Analysis (TGA-DTA) using Shimadzu DTG-60H. The NMW sample passing 150 μm was heated to 1000 °C with a heating rate of 10 °C/min under ambient environment.

### 2.3. Preparation of Slips

To prepare a slip for the production of green bodies, the dried NMW was slaked with water and mixed at a speed of 740 rpm using an improvised blunger Extreme Drill Press ETDP13 (EXTREME, Taiwan) and then left to age overnight. The aged slip was then wet screened at 150 μm to obtain a slip with particle size of less than 150 μm. The specific gravity of the obtained slip was then measured.

To determine the mass of the NMW in the slip, Brongniart's formula was used. The mass of the feldspar which act as a non-plastic filler was determined based on the composition of the target formulations. The mixture was thoroughly blended using the blunger press at a speed of 740 rpm for 10 min to attain homogeneity.

The fluidity of the slip was adjusted by the addition of concentrated sodium silicate of 2.92–3.09 wt% based on dry solid loading. The mixtures were left to age overnight for homogenization and stability. Casting properties of the aged slip for each of the formulation were measured such as the casting rate and the rheology that was determined at continuous stirring mode using a LVDVE 230 Brookfield viscometer (Brookfield Engineering Laboratories, Inc., Middleboro, MA, USA) with a spindle #6 at angular speed of 100 rpm for pure NMW and 30 rpm for formulated NMW slips. The density of the slips is within $1.70 \pm 0.09$.

### 2.4. Production of Green Bodies and Sintered Test Bars

The aged slips were thoroughly mixed using a blunger for 10 min and then casted into the test bar plaster mold. As the slip dries up in the mold, its level goes down and more slip was then added to replenish the slip. Refilling the mold with a slip is continued until a slight separation of the cast from the mold is observed, which indicates that the cast is ready for demolding. Each of the test bar green body is marked with 50 mm line as reference for the measurement of its shrinkage. The test bars were dried at 110 °C until a constant weight was attained. Sintering followed at 3 different firing temperatures—800 °C, 975 °C and 1050 °C—in an electric muffle furnace (SH Scientific, SH-FU-36MHSH Scientific Co., Ltd., Sejong, Korea).

The physical characteristics of the sintered test bars were determined in terms of the total linear shrinkage (TLS), which is the total shrinkage of drying and firing, loss on ignition (LOI), water absorption (WA), apparent porosity (AP) and modulus of rupture (MOR) that was measured using the universal testing machine Shimadzu AGS-X series with a maximum load capacity of 5000N. The equations used to obtain the properties mentioned can be found in published works [21–23].

## 3. Results and Discussion

This section covers the results of the characterization of the NMW properties that are relevant to the production of ceramic wall and floor tiles as well as the properties of the resulting ceramic products.

### 3.1. Raw Material Characterization

3.1.1. Particle Size Distribution

The PSD of the NMW is presented in Figure 1, and it is shown that 99.33% of the particles is less than 150 μm. This indicates that the NMW is already fine enough to qualify it as suitable raw material for ceramic products; thus, there is no need for further comminution. Eliminating comminution in the production of ceramic products using NMW marks a huge savings in energy, thereby significantly reducing the carbon footprint.

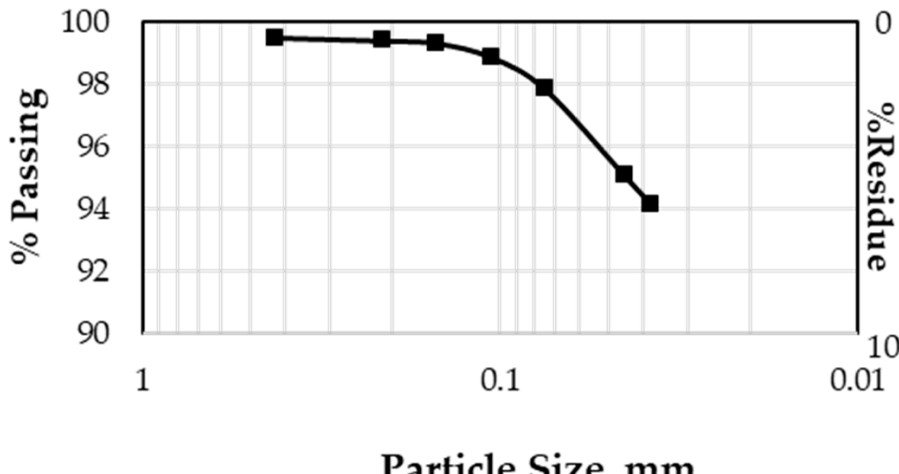

**Figure 1.** Particle size distribution (PSD) of nickel laterite mine waste (NMW).

3.1.2. Chemical and Mineralogical Characterization of NMW

The chemical composition of both NMW and feldspar primarily consists of iron oxide ($Fe_2O_3$), silica ($SiO_2$), alumina ($Al_2O_3$), magnesia (MgO), calcia (CaO) and nickel oxide (NiO), as shown in Table 1. NMW contains some minor to trace amounts of $K_2O$, MnO, $TiO_2$ and $Cr_2O_3$. Moreover, it has a relatively low alumina and silica contents and relatively high iron oxide ($Fe_2O_3$) at 46.26%. This amount of iron oxide in the NMW is comparable to that in the usual red clays used for the traditional ceramic production [24] and is even higher for those on terracotta production [25]. Thus, to understand how the slip derived from NMW will behave during the casting of ceramic products, further characterization must be performed, particularly on the oxide and mineral analyses. These analyses of the raw materials are necessary in the formulation of the slip to achieve successful casting and production of ceramic products. These minerals present in the NMW have inherent properties that will possibly influence the processing and thus the quality of product. The formulation of the slip for the production of the ceramic products is based on the quantity of the alumina, silica, fluxes and other oxides present in the raw material.

**Table 1.** Chemical composition of NMW and Feldspar.

| Mass % | $SiO_2$ | $Al_2O_3$ | $Fe_2O_3$ | MgO | CaO | NiO | $Cr_2O_3$ | MnO | $TiO_2$ | $Na_2O$ | $K_2O$ | SrO | $SO_3$ | $P_2O_5$ | LOI |
|---|---|---|---|---|---|---|---|---|---|---|---|---|---|---|---|
| NMW | 24.34 | 9.20 | 46.26 | 15.10 | 0.71 | 1.47 | 2.14 | 0.72 | 0.05 | | | | | | — |
| San Nicholas clay [25] | 53.7 | 16.9 | 12.9 | 4.72 | 1.79 | — | — | 0.12 | 1.09 | 1.8 | 0.35 | — | | 0.09 | 6.61 |
| Ma. Cristina Red Clay [24] | 29 | 18 | 46.33 | — | 1.38 | 0.06 | 0.13 | 0.62 | 3.13 | — | 0.08 | 0.05 | 0.6 | — | — |
| Feldspar | 60.65 | 12.59 | 0.31 | 5.16 | 21.12 | | | 0.02 | 0.15 | — | — | — | — | | |

Note: "—" means below detection limit.

Mineralogical analysis using XRD of the NMW shown in Figure 2 reveals that the minerals present are goethite, lizardite, spinel, quartz, magnetite, maghemite, hematite, montmorillonite, andesine, labradorite and anorthite where some of these minerals were also identified in the study of Longos et al. who used the same NMW source [26]. The summary of the minerals with their corresponding formula is presented in Table 2. The results of the XRF analysis of the iron content are consistent with the presence of the iron-bearing minerals such as goethite, magnetite, maghemite and hematite. Phyllosilicate minerals such as montmorillonite, lizardite and other silicates can be potentially useful for

slip casting production of ceramic tiles. However, this could also pose some challenges since the usual ceramic tile production generally uses kaolin and bentonite types of clays.

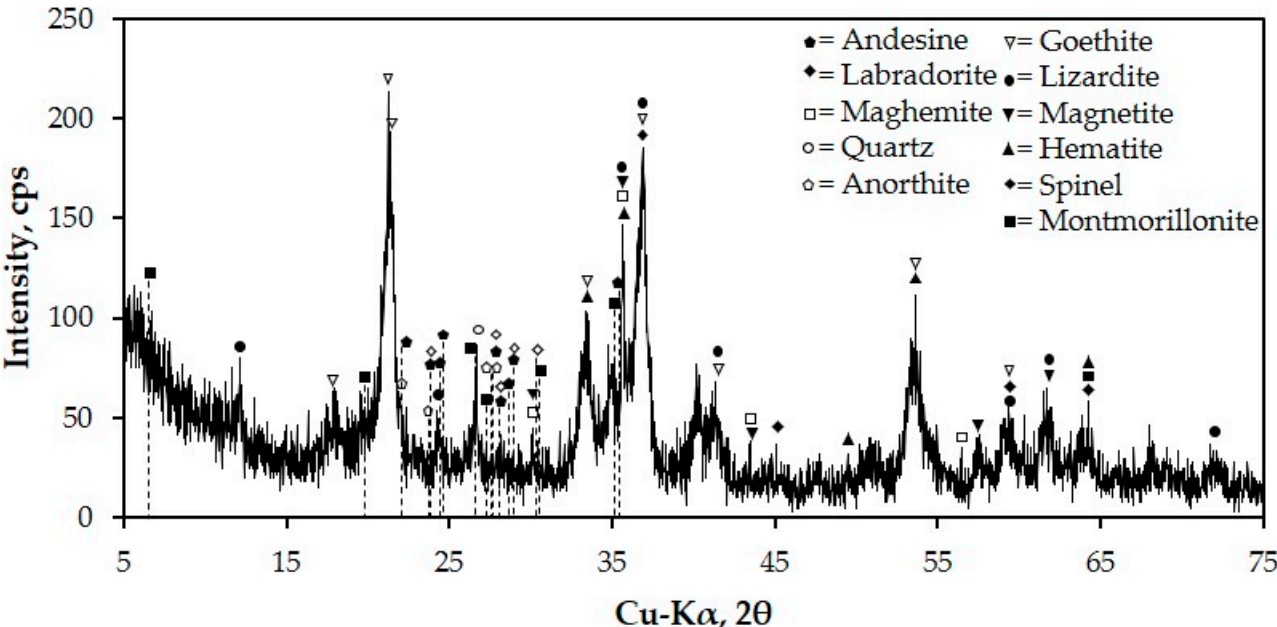

**Figure 2.** XRD pattern of the NMW sample.

**Table 2.** A list of identified minerals in the NMW sample.

| Mineral ID | Chemical Formula |
|---|---|
| Goethite | $Fe^{3+}O(OH)$ |
| Lizardite | $Mg_3Si_2O_5(OH)_4$ |
| Spinel | $MgAl_2O_4$ |
| Quartz | $SiO_2$ |
| Magnetite | $Fe_3O_4$ |
| Maghemite | $\gamma\text{-}Fe_2O_3$ |
| Hematite | $Fe_2O_3$ |
| Montmorillonite | $(Na, Ca)_{0.3} (Al, Mg)_2 Si_4 O_{10} (OH)_2 \bullet n\ H_2O$ |
| Andesine | $(Na, Ca) (Si, Al)_4 O_8$ |
| Labradorite | $(Ca, Na)(Si, Al)_4 O_8$ |
| Anorthite | $CaAl_2Si_2O_8$ |

Furthermore, the NMW of the same site was investigated in a previous study for its possible ability to release hazardous elements by toxicity characteristic leaching procedure (TCLP) [27–29]. Based on the TCLP results, the leaching concentrations of chromium, cadmium, arsenic, lead and selenium were found to be below the United States Environmental Protection Agency limits and Philippines' DENR Administrative Order 2013-22 Standards [26]. This could mean that NMW is non-hazardous.

### 3.1.3. Thermal Stability Test of NMW

The thermal stability of the NMW was studied from the characteristic TG-DTA thermograms presented in Figure 3. It can be seen that the total mass loss of the NMW silt is 16.75%. The DTA pattern shows four regions of DTA peaks, three of which correspond to endothermic reactions and one corresponds to exothermic reaction. The first region shows an endothermic peak at 91.10 °C with mass loss of 4.95%, which can be attributed to the elimination of free water based on the TG curve. The second region presents another endothermic peak at 295.34 °C with 7.79% mass loss, which is caused by the removal of crystal water. The third region is another endothermic reaction at 637.54 °C with mass

loss of 2.95%, which indicates the loss of hydroxyl group. The last region is an exothermic reaction at 818.32 °C with 1.06% mass loss due to the decomposition of silicate such as montmorillonite and lizardite [30–32].

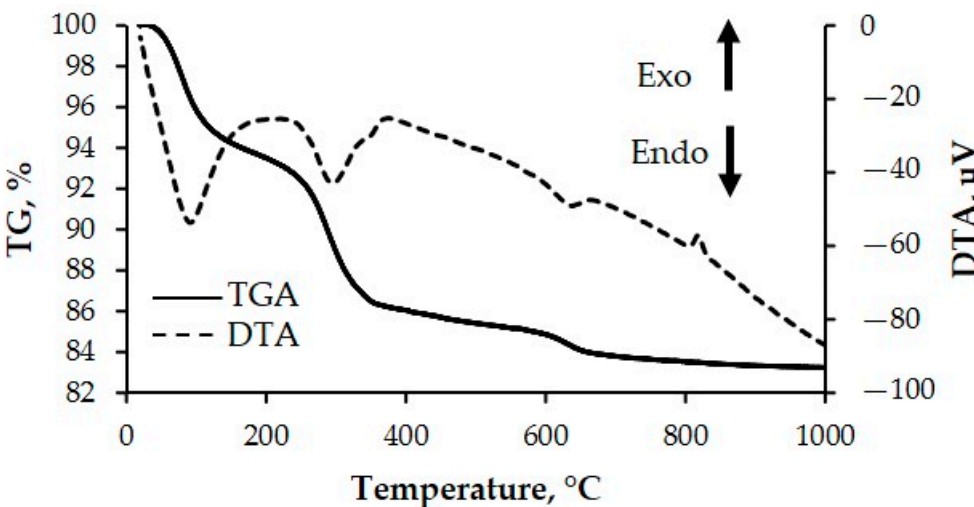

**Figure 3.** TGA-DTA thermograms of NMW sample.

### 3.1.4. Atterberg Limits

The Atterberg limits of the NMW shows a liquid limit of 74% and plastic limit of 52% which resulted in a plasticity index of 22%. This indicates that the nickel laterite silt from NMW is classified as high plasticity clay [19,20]. Figure 4 shows a plot of the plasticity of NMW. The "A" line in the chart is the empirical boundary that separates inorganic clays from inorganic silts and organic soils [33,34]. The clays are plotted above the "A" line, whereas the silts are generally plotted below it. However, there are exceptions because some clays contain organic content. These organic clays are plotted below the "A" line along with the silts. Table 3 shows the summary of the classification of soil based on ASTM D2487 [35] that is close to the region of classification of NMW. NMW is classified as an elastic silt (MH) in accordance with ASTM D2487 [35], stating that elastic soil has a liquid limit of 50 or greater. This means that NMW is suitable for brick tile production by plastic mass forming or hand wedging since the range of plasticity index is within 8–25 [33]. Furthermore, the plastic mass of pure NMW was found to have higher plasticity index and liquid limit than bricks from river silt [20]. However, in this study, a slip casting method was employed for forming tiles to explore the versatility of NMW to cast tiles with surface features, simulating casting of ceramics with complex surface features or shapes. The surface features could include embossed grid lines for mounting purposes and embossed design for aesthetic purposes.

### 3.2. Characterization of the Formulated Slips

Five formulations were prepared for the evaluation of NMW for ceramic tiles as shown in Table 4. US contains 100% NMW for tile production. Formulations USF1 to USF5 contain decreasing amounts of NMW added with feldspar. To simulate the stability of slip during slip casting, the viscosity of slips with respect to time at continuous stirring mode under a given constant shear rate was determined. The stirring rate was set to 100 rpm ($14.2\,\mathrm{s}^{-1}$) for pure NMW slip and 30 rpm ($21.4\,\mathrm{s}^{-1}$) for NMW formulated slips. Initial study shows a high stirring rate of 100 rpm for the pure nickel laterite mine waste was found to be stable inorder for its viscosity to be determined by the viscometer. However, the viscosity of formulated slips can already be detected even at low intensities of agitation or low stirring rate at 30 rpm. As shown in Figure 5, the viscosity of the formulated slips decreases with decreasing NMW content in the formulation. Most formulated slips showed a slightly stable viscosity over time such as USF1, USF2, USF3 and USF4 except for USF5, which

exhibits rheopectic behavior. Rheopectic behavior is the condition where a slight increase in viscosity over time (shear thickening) is observed even under an imposed constant shearing action. During the continued stirring of USF5 slip, a preferential buildup of structure in the micron sized particles was observed, and shear-induced crystallization may be responsible for this behavior [36]. However, despite the slight increase in the viscosity of USF5 over time, it exhibits the lowest viscosity among the formulated slips. Moreover, USF4 exhibits the most stable viscosity and shows only a slightly higher viscosity than that of USF5 within 60 min of continuous stirring. This indicates that both USF4 and USF5 may require only low energy for continuous stirring requirements to maintain its fluidity during casting.

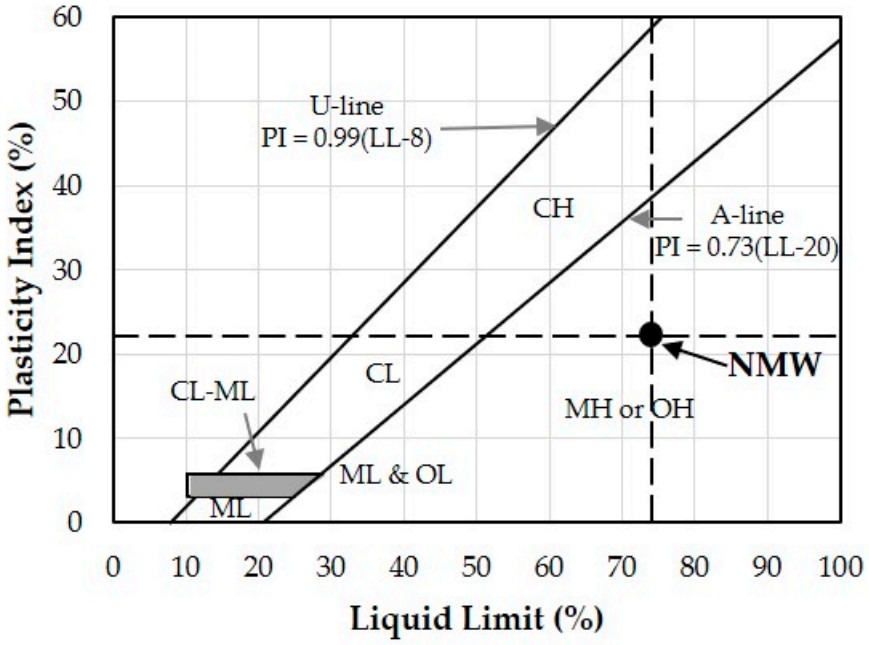

**Figure 4.** Plasticity Index and Liquid Limit of NMW on the Plasticity chart [34].

**Table 3.** Summary of classification of soil based on ASTM D 2487 [35] for NMW.

| Code | Type of Soil | PI | LL | Position (Basis: "A" Line) |
|---|---|---|---|---|
| CL-ML | Silty clay | 4–7 | | On or above |
| ML | Silt | | <50 | Below |
| MH | Elastic Silt | | ≥50 | Below |
| OL | Organic clay | ≥4 | <50 | On or above |
| | Organic silt | <4 | | Below |
| OH | Organic clay | | ≥50 | On or above |
| | Organic silt | | | Below |

**Table 4.** Formulation of NMW tiles.

| Formulation | Empirical Formula |
|---|---|
| USF1 | 4.11 MgO• 0.18 CaO• $Al_2O_3$• 4.55 $SiO_2$• 3.17 $Fe_2O_3$ |
| USF2 | 4.03 MgO• 0.26 CaO• $Al_2O_3$• 4.65 $SiO_2$• 3.08 $Fe_2O_3$ |
| USF3 | 3.40 MgO• 0.85 CaO• $Al_2O_3$• 5.39 $SiO_2$• 2.44 $Fe_2O_3$ |
| USF4 | 2.77 MgO• 1.44 CaO• $Al_2O_3$• 6.14 $SiO_2$• 1.79 $Fe_2O_3$ |
| USF5 | 1.97 MgO• 2.18 CaO• $Al_2O_3$• 7.08 $SiO_2$• 0.98 $Fe_2O_3$ |

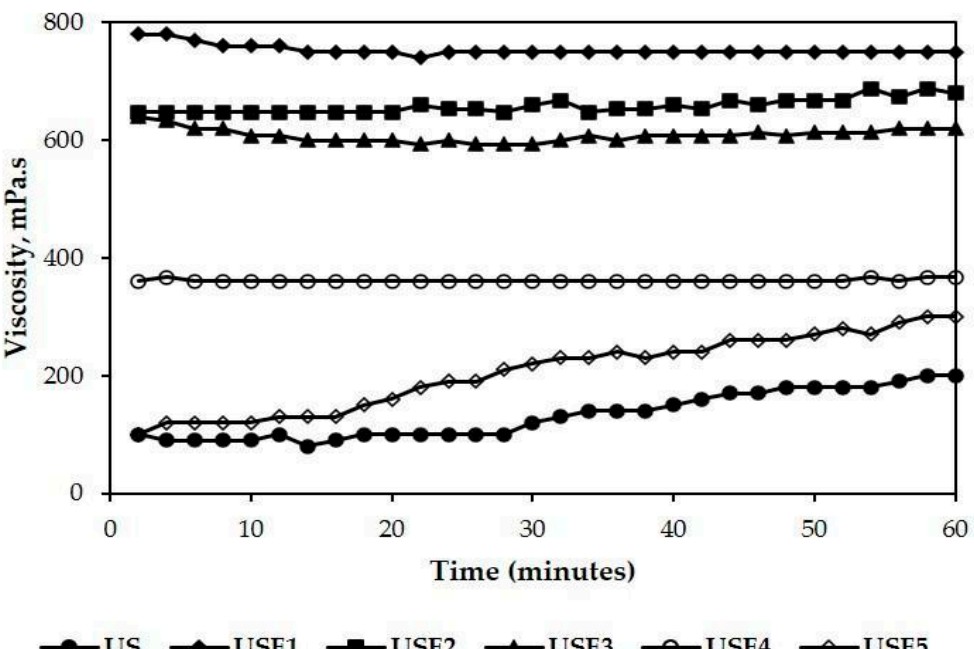

**Figure 5.** A plot of viscosity with time at 14.2 s$^{-1}$ (100 rpm for pure NMW slip) and 21.4 s$^{-1}$ (30 rpm for formulated NMW slips); 29.5 °C.

A comparison of the draining time, demolding time, casting time, wet thickness and casting rate of the pure NMW slip and the different formulations of slips is presented in Figure 6. Both pure and formulated slips yielded wet cast thickness ranging from 5.17 to 7 mm, which indicate that they can be casted to produce tiles with a range of thickness between 10 and 14 mm. The casting rate estimates are close to each other but cannot be used to determine which slips are best for tile casting. However, among the five formulations, USF4 and USF5 exhibit the shortest draining, demolding and casting time, at least 35 min, for casting a 10 mm thick tile. For a 14 mm thick tile, casting took 80–125 min, which means that casting a 10 mm thick tile is more favorably economical.

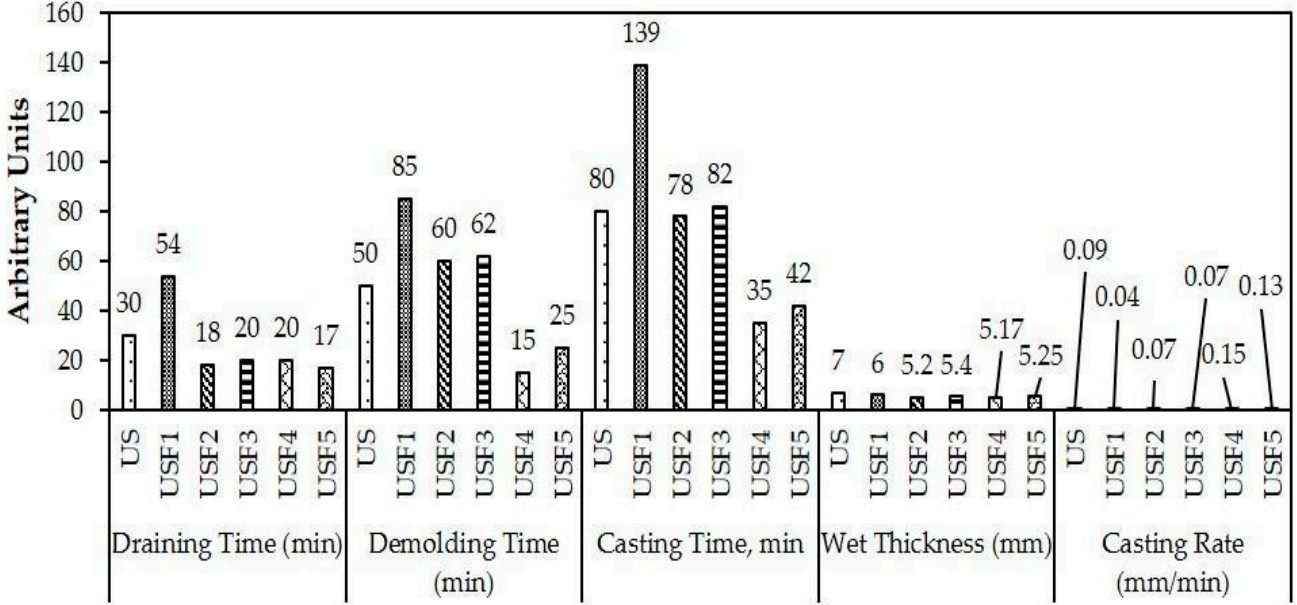

**Figure 6.** Comparison of casting properties of the different slip formulations.

### 3.3. Physical and Mechanical Characterization of the Ceramic Tiles

Physical and mechanical characterization of the ceramic tiles used test bars produced from the different slip formulations. USF1 and USF2, which were characterized by a high casting time, obtained flabby casts, which resulted in hollow rather than solid test bars. Cracks were easily induced on the casts during demolding, as shown in Figure 7(a1). Hence, high casting time does not really guarantee good casts. USF3 on the other hand produced a slightly flabby cast wherein the test bars were still formed; however, it generally failed in the casting of larger tiles due to the formation of cracks during demolding as shown in Figure 7(a2). The flabby casts obtained from the casting of USF1, USF2 and USF3 are consistent with their high viscosity results. Hence, no data were obtained further for the physical properties of USF1, USF2 and USF3. Successful casting and firing at 1050 °C of ceramic tiles, as shown in Figure 7c, were obtained from USF4 and USF5, which have both relatively low NMW content.

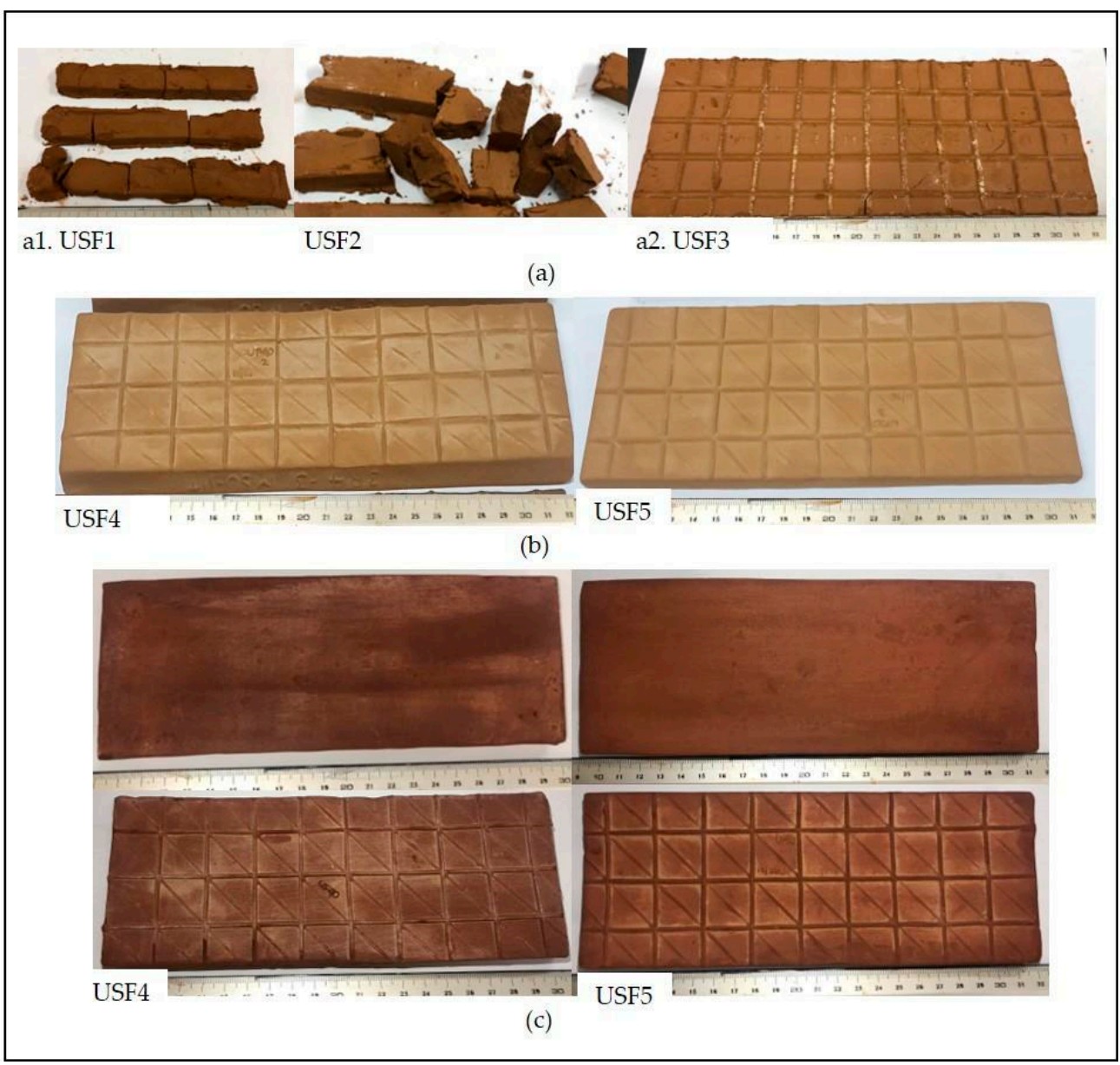

**Figure 7.** Tiles from NMW slip. (**a**) Failed casting using USF1, USF2 and USF3; (**b**) successful casting of tiles using USF4 and USF5; and (**c**) USF4 and USF5 tiles fired at 1050 °C.

The average results of the total linear shrinkage, loss on ignition, water adsorption and apparent porosity of the tiles produced from USF4 and USF5 at three different firing temperatures are presented in Figure 8. The total shrinkage is generally lower at low temperatures, as shown in Figure 8a. These values of total shrinkage are necessary to forecast the desired size of the fired tiles during the making of the plaster mold for casting tiles. Usually, a low value of shrinkage is favorable to achieve dimension of the wet cast tile close to the dimension of the final tile. However, other properties should be considered in choosing the best formulation. All tiles with NMW have higher shrinkage than bricks from river silt [20] and dredging sediment [37] since river silt and dredging sediment are highly siliceous while NMW has low silica content. The loss on ignition of the formulated NMW as shown in Figure 8b is lower than the mass loss obtained from TGA (16.75%). This means that the feldspar has a negligible contribution to the mass loss of the formulated fired tiles which could be attributed solely to the NMW content. Both USF4 and USF5 could have undergone initial vitrification as early as 900 °C [38], which compacted the clay matrix at high temperature (1050 °C) and aided in porosity reduction. During vitrification, liquid phases of clays and feldspars are starting to form and fill the matrix, which promote its densification and contraction of the interior structure due to the rearrangement of particles induced by capillarity and surface tension [39]. USF4 has higher flux (CaO, MgO)/silica ratio or higher (CaO, MgO) content, relative to silica, compared to USF5. CaO and MgO are typical oxides that can lower the melting point of silica, which forms into a liquid during firing. This liquid begins to fuse filling up the pore spaces in the tile matrix, which decreases porosity as well as the water absorption, due to the formation of this liquid during firing or its vitirification [40], as evident in water absorption and apparent porosity in Figure 8c,d, respectively. This is also accompanied by a higher shrinkage of USF4 than of USF5, as shown in Figure 8a. CaO and MgO are primarily contributed by feldspar and NMW, respectively. This shows the high feldspar content of both USF4 and USF5 successfully lowered both the water absorption and apparent porosity particularly at high firing temperatures (975 °C and 1050 °C). However, tiles of USF4 and USF5 fired at 1050 °C have higher water absorption, resulting in a lower MOR than the bricks with river silt [20] and tiles with dredging sediment [37].

The modulus of rupture at different temperatures is presented in Figure 9. Tiles with high MOR has low water absorption and low apparent porosity. Test bars fired at 975 and 1050 °C have higher MOR values than those fired at low temperature (800 °C) [38]. This could be due to the presence of liquid phases filling the matrix formed during vitrification as early as 900 °C [38]. The lower MOR value obtained at 975 °C than at 1050 °C could be attributed to the incomplete structural transformation resulting in the heterogeneity of phases such as silica carrying spinel-type structures [38]. USF4 has higher MOR (15.31 MPa) than USF5 (11.86 MPa) when fired at 1050 °C, which concurs from USF4 having lower water absorption and lower apparent porosity than USF5. This could mean that USF4 have more solid phases than USF5 in the tile matrix resisting the applied load during MOR test prior to their fracture. In this study, the production of NMW tiles with formulation of USF4 fired at 1050 °C obtained the highest MOR of 15.31 MPa.

### 3.4. Quality classification

The summary of the quality compliance for the tiles is shown in Table 5. Both tiles from USF4 and USF5 were evaluated with respect to four standards: Philippine National Standard (PNS) [41], Chinese National Standard (CNS) [42], Indian Council of Ceramic Tiles and Sanitaryware (ICCTAS) [43] and ISO standard 13006 [20]. All tiles fired at different temperatures passed the water absorption requirement for CNS type III (50%) and ISO standard 13006 type AIII (>10%). Both USF4 and USF5 fired at 1050 °C passed the modulus of rupture for ISO standard 13006 type AIII (>8 MPa) and AII$_{b-2}$ (>9 MPa). Furthermore, only USF4 fired at 1050 °C passed the PNS requirement ($\geq$15 N/mm$^2$) for wall tiles, ICCTAS requirement for wall tile ($\geq$15 N/mm$^2$) and ISO standard 130006 for type AII$_{a-2}$ (>13 MPa).

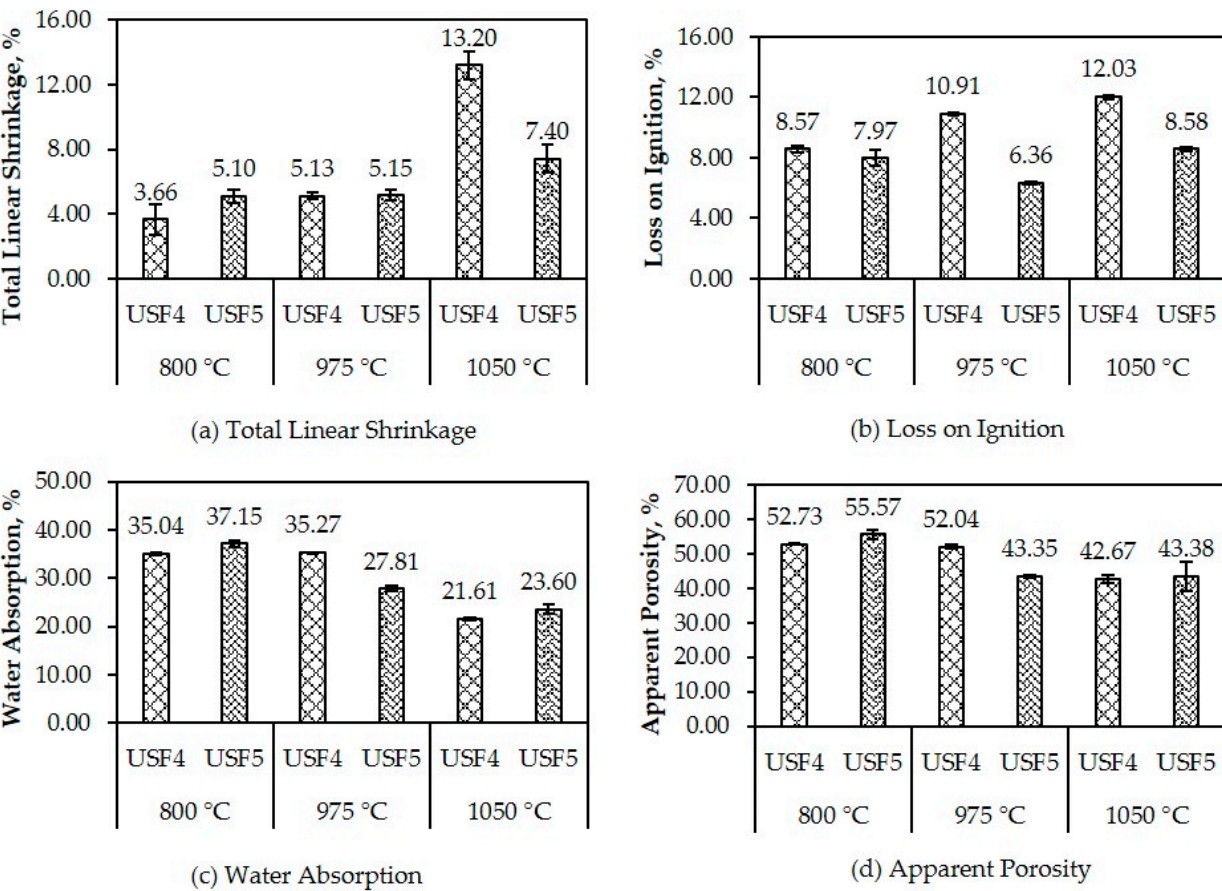

**Figure 8.** Results on (**a**) shrinkage, (**b**) loss on ignition (LOI), (**c**) water absorption and (**d**) apparent porosity at different temperature.

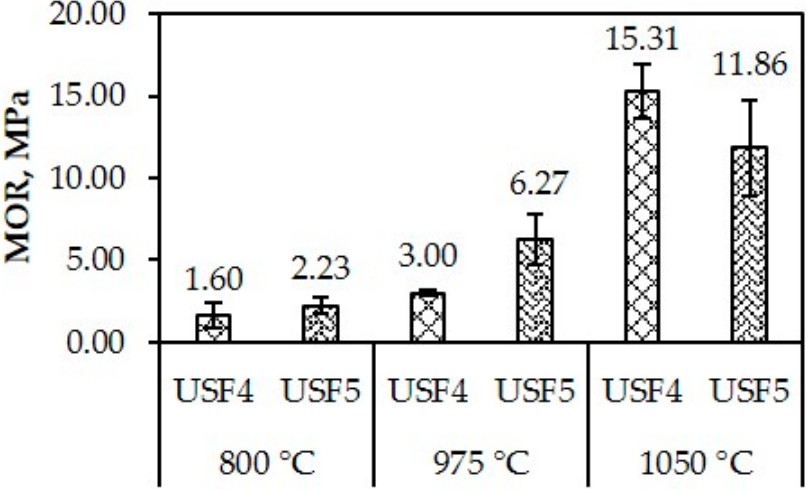

**Figure 9.** Results on modulus of rupture at different sintering temperature.

This study showed that Nickel laterite mine waste could be utilized as raw materials for both wall and floor tiles across various standards. Feldspar could be added to enhance casting to form NMW tiles. In terms of relative amount of valorization of waste, 10 wt% and 12 wt% were utilized for tiles with river silt and bricks with dredging sediment, respectively. These values are lower compared to the tiles with NMW, which employed at least three-fold their amount. This could add value to the voluminous amount of NMW than just being stored in many siltation ponds that consistently require high management

costs of siltation ponds by mining industries. For tile production, the utilization of NMW does not require further comminution, which reduces the carbon footprint in ceramic tile production. At least 200 g of NMW will be used for each tile, and this depends on the formulation. Thus, for a typical indoor floor for a 30 m$^2$ living quarter, at least 1150 tiles or 230 kg (approximately 0.0875 m$^3$) of NMW are needed for tile production to cover the floor area. This shows that the amount of silt obtained in 2020 for the aforementioned siltation pond is more than enough to produce the ceramic wall and floor tiles.

**Table 5.** Quality compliance summary.

| Formulation | Temperature | Judgement Criteria | | |
| --- | --- | --- | --- | --- |
| | | NMW Tile | | |
| | | Modulus of Rupture, MPa | Water Absorption, % | Remarks |
| USF4 | 800 °C | 2.23 ± 0.51 | 35.04 ± 0.32 | ●▲ |
| | 975 °C | 3.00 ± 0.18 | 35.27 ± 0.08 | ●▲ |
| | 1050 °C | 15.31± 1.66 | 21.61 ± 0.34 | ●▲◇□△○⬡ |
| USF5 | 800 °C | 1.60 ± 0.77 | 37.15 ± 0.71 | ●▲ |
| | 975 °C | 6.27 ± 1.52 | 27.81± 0.48 | ●▲ |
| | 1050 °C | 11.86 ± 2.92 | 23.60 ± 0.99 | ●▲△⬡ |

●—Water absorption passed for CNS Type III floor tile. ◇—Modulus of Rupture passed for ICCTAS wall tile. □—Modulus of Rupture passed for PNS wall tile. ▲—Water absoprtion passed for ISO Standard 13006 AIII. △—Modulus of Rupture passed for ISO Standard 13006 AIII. ○—Modulus of Rupture passed for ISO Standard 13006 AII$_{a-2}$. ⬡—Modulus of Rupture passed for ISO Standard 13006 AII$_{b-2}$.

## 4. Conclusions

The study makes a pioneering attempt to produce ceramic tiles from NMW as full replacements for the clay content such as kaolin, bentonites and typical red clay by solid slip casting method. This study has yielded some promising results, and its feasible application is advised. Repurposing mine wastes provides dual environmental and economic benefits, as it avoids further waste disposal costs while also reducing the usage of ceramic raw materials. Based on the experimental results of the study, the following conclusions are drawn:

- Nickel laterite mine waste is composed of very fine particles with 94.16% 38 micron particles.
- The chemical composition of NMW reveals that it has low alumina and silica contents but with high iron oxide content of 46.26%. The results coincide with XRD results wherein most of the minerals contain Fe such as goethite, maghemite, magnetite and hematite. It also contains phyllosilicates such as montmorillonite, lizardite and other silicates that have potential for slip casting production.
- NMW is classified as high plasticity clay with respect to its Atterberg limits, which is suitable for brick tile production.
- The viscosity of pure and formulated slips decreases with decreasing NMW content in the formulation with USF5 having the least viscosity. Among the formulations, USF4 showed stable viscosity along time. Casting properties showed that formulated slips have cast thickness ranging from 5 to 7 mm and, therefore, can cast a thick tile about 10–14 mm.
- Tiles were produced from USF4 and USF5 only. Physical properties showed total linear shrinkage, loss on ignition, water absorption and apparent porosity were generally low at low temperatures. Both USF4 and USF5 had initial vitrification at temperatures as early as 975 °C, which compacted the clay matrix at 1050 °C. USF4 and USF5 fired at 1050 °C had the highest MOR.



- Both USF4 and USF5 passed the CNS Type III water absorption requirement for floor tiles and ISO standard 13006 type AIII for water absorption for the three firing temperatures. Both USF4 and USF5 fired at 1050 °C passed the modulus of rupture for ISO standard 13006 type AIII and $AII_{b-2}$. At 1050 °C, only USF4 passed the PNS MOR requirement for wall tile, the ICCTAS MOR requirement for wall tile and ISO standard 130006 MOR requirement for type $AII_{a-2}$.

**Author Contributions:** Conceptualization, I.C.B.-A.; methodology, I.C.B.-A., F.J.A.E., R.H.L.A. and V.J.T.R.; validation, C.B.T., V.J.T.R., R.V.R.V. and I.C.B.-A.; formal analysis, I.C.B.-A., F.J.A.E., C.B.T., R.V.R.V. and V.J.T.R.; investigation, I.C.B.-A., R.H.L.A., F.J.A.E. and V.J.T.R.; resources, I.C.B.-A. and V.J.T.R.; data curation, F.J.A.E.; writing—original draft preparation, F.J.A.E., C.B.T., V.J.T.R. and I.C.B.-A.; writing—review and editing, F.J.A.E., C.B.T., V.J.T.R., R.V.R.V. and I.C.B.-A.; visualization F.J.A.E., I.C.B.-A.,C.B.T., R.V.R.V. and V.J.T.R.; supervision, C.B.T., V.J.T.R. and I.C.B.-A.; project administration, I.C.B.-A.; funding acquisition, I.C.B.-A. and V.J.T.R. All authors have read and agreed to the published version of the manuscript.

**Funding:** This research was funded by The Department of Science and Technology—Philippine Council for Industry, Energy and Emerging Technology Research and Development—grant number 7131—under the implementing agency of the College of Engineering and Technology, Mindanao State University—Iligan Institute of Technology, Philippines.

**Acknowledgments:** Authors acknowledge the following organizations: Agata Mining Ventures, Inc. (AMVI), Ceramic Engineering and Department of Materials and Resources Engineering and Technology of Mindanao State University—Iligan Institute of Technology. Acknowledgement is also extended to Engr. Aljon Rey A. Lagayada, Engr. Alyssa S. Rabadon and Engr. Joel O. Esencia for their administrative and technical support.

**Conflicts of Interest:** The authors declare no conflict of interest. The funders had no role in the design of the study; in the collection, analyses, or interpretation of data; in the writing of the manuscript; or in the decision to publish the results.

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
