# Peer review of "Development of Ceramic Tiles from Philippine Nickel Laterite Mine Waste by Ceramic Casting Method"

_minerals, doi:10.3390/min12050579_

Round 1
Reviewer 1 Report
The percentages of quartz and lizardite in XRD are minor, therefore, they do not correspond with the SiO2 content in the chemical analysis by XRF (15.6%), this may mean that the diffraction peak near 32 degrees corresponds to feldspar rather than iron oxides.
Also, the amount of lizardite and spinel is overestimated in the XRD analysis, which is verified by the absence of Mg in the chemical analysis.
Put the PDFs in Fig. 2 or in Table 2.
It is recommended to index again since in a single peak there are 4 different species, or to explain why these overlaps exist.
It would be convenient that the authors present the chemical formulas of the final solution.
It would be convenient that the authors show the tables of the values obtained from the characteristic toxicity leaching procedure (TCLP).
To make a more concrete conclusion and this information could be used in the discussion.
It would be interesting to present a comparative of their product to those elaborated in a traditional way both in physical and chemical properties.
It is recommended to consult some articles such as
Title Production Using Wastes From Minimg Industry Of The Mining Dristict Pachuca Real Delmonte, Tms (The Minerals, Metals & Materials Society), Pp 203-209, (2012).
Design And Production Of A New Construction Material (Bricks), Using Mining Tailings, International Journal Of Engineering Sciences & Research Technology, (2017).
Author Response
"Please see the attachment." Thank you.

Reviewer 2 Report
The paper deals with an interesting topic, the development of ceramic tiles from Philippine nickel laterite mine waste. Overall, the manuscript is well structured and the authors reported a wide and detailed characterization. Nevertheless, there are some points that the authors need to address.
- Any reasons why the nickel laterite mine waste is currently not utilized to extract iron due to its high iron oxide content at 73.11%.
- Please describe the abbreviations in Figure 4. Authors should elaborate on the reference [31] in the text.
- Please provide the formulations of US, USF1, USF2, USF3, USF4, and USF5.
- There is a lack of discussions explicitly addressing how the properties of the fabricated tiles relate to the formulation of the tiles.
- Authors should perform XRD analysis on the fabricated tiles, which can give some explanations on the properties of the tiles.
- Please provide error bars for Figure 8 and Figure 9.
Author Response
"Please see attachment." Thank you.

Reviewer 3 Report
Dear Authors,
Dear Authors, the manuscript submitted for publication in Minerals (1684130) is a study that reports the feasibility to valorise NMW in the ceramic tiles production.
The study is well developed and the results are obtained by using different techniques (XRD, TG/DTA, rheological misures, etc) and other characterization following ASTM rules.
The Tables and Figures are clear and the number is right. The references collected are numerous and meet the argument.
In my opinion, the research reported in the manuscript is interesting but it must be improved before publication, for example: a) the authors do not report a Table with a batch composition prepared in order to understand the amount of NMW involved in the formulations; b) missing the XRD analysis of the obtained products in order to discuss the reactivity of NMW with the ceramic raw materials during the firing step; c)based on a cited reference these residues should not be dangerous even if the chemical analysis reports the presence of chromium among other metals. There is no leaching test, Justify.
In the file attached, the authors can find some corrections and suggestions to improve the text .

Author Response
"Please see the attachment." Thank you.

Round 2
Reviewer 1 Report
the authors complied with the recommendations
Reviewer 2 Report
The manuscript can be considered for publication